# DAT-MT Accelerated Graph Fusion Dependency Parsing Model for Small Samples in Professional Fields

**DOI:** 10.3390/e25101444

**Published:** 2023-10-12

**Authors:** Rui Li, Shili Shu, Shunli Wang, Yang Liu, Yanhao Li, Mingjun Peng

**Affiliations:** 1State Key Laboratory of Information Engineering in Surveying, Mapping and Remote Sensing, Wuhan University, Wuhan 430072, China; ruili@whu.edu.cn (R.L.); shilishu@whu.edu.cn (S.S.); shunliwang@whu.edu.cn (S.W.); yang_liu@whu.edu.cn (Y.L.); yanhao@whu.edu.cn (Y.L.); 2Wuhan Geomatics Institute, Wuhan 430079, China

**Keywords:** small sample, specific professional fields, graph fusion, double-array trie, multi-threading, dependency parsing

## Abstract

The rapid development of information technology has made the amount of information in massive texts far exceed human intuitive cognition, and dependency parsing can effectively deal with information overload. In the background of domain specialization, the migration and application of syntactic treebanks and the speed improvement in syntactic analysis models become the key to the efficiency of syntactic analysis. To realize domain migration of syntactic tree library and improve the speed of text parsing, this paper proposes a novel approach—the Double-Array Trie and Multi-threading (DAT-MT) accelerated graph fusion dependency parsing model. It effectively combines the specialized syntactic features from small-scale professional field corpus with the generalized syntactic features from large-scale news corpus, which improves the accuracy of syntactic relation recognition. Aiming at the problem of high space and time complexity brought by the graph fusion model, the DAT-MT method is proposed. It realizes the rapid mapping of massive Chinese character features to the model’s prior parameters and the parallel processing of calculation, thereby improving the parsing speed. The experimental results show that the unlabeled attachment score (UAS) and the labeled attachment score (LAS) of the model are improved by 13.34% and 14.82% compared with the model with only the professional field corpus and improved by 3.14% and 3.40% compared with the model only with news corpus; both indicators are better than DDParser and LTP 4 methods based on deep learning. Additionally, the method in this paper achieves a speedup of about 3.7 times compared to the method with a red-black tree index and a single thread. Efficient and accurate syntactic analysis methods will benefit the real-time processing of massive texts in professional fields, such as multi-dimensional semantic correlation, professional feature extraction, and domain knowledge graph construction.

## 1. Introduction

Against the backdrop of rapid development in technologies such as artificial intelligence and spatiotemporal big data, the increasingly pressing issue of “information overload” poses a challenge to the rapid and accurate extraction of the truly needed information for individuals [1,2]. Dependency syntax trees reveal the structure and grammatical relationships between words in a sentence, serving as a crucial semantic carrier of textual information and providing a foundation for other natural language processing tasks such as sentiment analysis, named entity recognition, and semantic similarity calculation [3,4,5]. In the trend towards domain specialization [6,7], while general-purpose syntax tree models can capture the structure and grammatical information of sentences in natural language processing, they lack domain-specific expertise [8,9]. By combining general-purpose syntax trees with specialized domain-specific applications, models can adapt to the terminology, expressions, and text structures of specific fields [10,11,12]. This can provide more accurate and efficient text analysis, information extraction, and intelligent question answering for relevant domains, helping to better handle and understand large amounts of domain-specific textual data [13,14,15].

Informatization and intelligence in professional fields mostly require powerful capabilities of information extraction and analysis, which can be summarized as the characteristics of professionalism, high accuracy, and high timeliness. However, in professional fields, there is often a lack of large-scale labeled sample data, and the labeling work is time consuming and labor intensive [16,17], resulting in low accuracy and low practicability of directly applying dependency syntax model to domain text data. At the same time, there are often specific language expression styles and forms in professional fields, and the dependency syntax model established with open-text data is often ineffective. Furthermore, the method based on the syntactic dependency model has higher space complexity and time complexity in the step of dependency syntax tree generation [18], which makes simple indexing and single-threaded computing unable to meet the high timeliness in professional fields. 

Therefore, it is important and necessary to construct a dependent syntactic model that is not only suitable for small sample text data in the professional field but also able to improve the computational efficiency, meeting the requirements of massive text data processing and analysis. The model will be conducive to the rapid syntactic analysis of texts in professional fields, support the extraction of professional elements and the construction of domain knowledge maps, and promote the intelligent development of professional fields. 

In this study, we first establish two dependency syntactic models (respectively called the element model and the news model) based on a small sample professional corpus and a large sample news corpus, respectively. Then, we propose a graph fusion model with the connection edge weight optimization method. The model can learn both specialized semantic features from the element model and general written semantic features from the news model. Meanwhile, in order to improve the speed of model fusion, we propose the DAT-MT method based on the Double-Array Trie and Multi-Threading. The flowchart is shown in Figure 1.

The main contributions of this study can be summarized as follows:According to the maximum spanning tree theory of dependency syntax, we designed a graph fusion model based on the connection edge weight optimization method. It effectively integrates the colloquial and specialized syntactic features in the small sample professional field texts with the common written expression features in the large sample news texts, which improves the accuracy of the model’s syntactic structure recognition in the professional field.We propose a new DAT-MT method to realize high-performance parsing of graph fusion model to meet massive text processing requirements. Our experimental results show that the processing speed of the DAT-MT method is much faster than the one of a single-threaded and red-black tree.

The remainder of this paper is organized as follows. In Section 2, we provide an overview of related research in the field. In Section 3, we introduce the methods proposed in this study, which are the dependency parsing model construction of a single treebank, the fusion model construction based on two single treebank models, and the DAT-MT methods. In Section 4, we present the experimental results focusing on accuracy and speed. Finally, in Section 5, we draw conclusions from our findings and offer suggestions for future research directions.

## 2. Related Work

Dependency parsing is an important task in natural language processing that addresses the fundamental question about the semantics of a sentence: “[who] [did what] [to whom]” [19,20]. Its main objective is to determine the dependency relationships between words in a sentence. With the emergence of large-scale annotated treebanks [21], statistically based parsing methods have been widely applied, and the transfer of syntactic treebanks to specialized domains and the improvement in parsing model efficiency have gradually become areas of research interest [22,23].

### 2.1. Domain Migration of Syntactic Tree Library

Syntactic treebanks serve as the foundation for building dependency parsing models, providing reliable training and testing datasets for syntactic analysis models. However, currently, syntactic dependency treebanks built on news corpora have low domain portability and are not suitable for highly specialized domains [24,25]. To address this issue, Sano et al. [26] transformed and unified the representation of various treebank types, such as Stanford structure and Penn structure, so as to fuse treebank or transform treebank at the data level. Yu et al. [27] constructed a domain-specific treebank for specific applications and specialized corpus to complete dependency parsing tasks of corresponding domains. Some researchers [28,29] improved the accuracy of syntactic analysis in a specific domain by constructing a library of graph data structures, achieving the goal of treebank domain transplantation. Wu et al. [30] extracted cross information by comprehensively utilizing the dependency skeleton resolved by different dependency parsing parsers, thereby integrating dependency parsing models. Liu et al. [31] proposed a syntax and domain-aware model for program translation, which leveraged the syntax structure and domain knowledge to enhance the cross-lingual transfer ability.

According to existing research, although the scale of the treebank can be expanded by transforming and merging various syntactic treebanks, the professional characteristics of the recorded corpus cannot be effectively reflected in the treebank due to the differences in application fields. In the construction of a domain-specific treebank, firstly, it should be noted that there are little labeled data, and labeling is a costly and time-consuming task. Secondly, the specialized treebank with a small-scale corpus cannot contain complex syntactic information, which can easily cause the syntactic tree to generate incorrect results. As for model fusion, existing research mainly focuses on fusing treebanks with similar performance and more balanced samples.

### 2.2. Efficiency Improvement in the Syntax Analysis Model

The efficiency improvement in syntactic parsing models is crucial due to the challenges and demands posed by large-scale data processing, real-time language processing, resource consumption, and user experience [32,33]. However, models used for dependency syntactic parsing typically have large parameter sizes and complex computational structures, requiring more time and computational resources to complete parsing tasks [34,35]. Dependency parsing is a basic task of natural language processing. However, due to the large size, the parsing efficiency of the model is low, which makes it unable to meet the needs of real-time text processing and massive data syntactic analysis. In terms of improving the parsing speed of dependency syntactic models, Li et al. [36] realized the acceleration of the combined model of word segmentation and dependency syntactic analysis with a little loss of accuracy based on the over-training method. Goldberg and Elhadad [37] reduced the time consumption of model traversal to speed up the model analysis stage by aligning the position of features and weights in the model. Loglo [38] improved the performance of the dependency parsing model by establishing the same data organization model as the machine learning model and introducing pre-statistical information. Anderson and Gómez-Rodríguez [39] used teacher–student distillation to improve the efficiency of the Biaffine dependency parser, which achieved high performance in terms of accuracy and parsing speed.

In terms of improving the analysis speed of dependency parsing models, current research mainly focuses on improving the time-consuming model analysis by optimizing the model structure but does not make full use of the current ubiquitous multi-core computing resources. Therefore, on the basis of the public treebank construction research, we will effectively integrate the features of small-sample domain-specific treebanks. Furthermore, the parallel computing method is used to improve the speed of the fused dependency parsing model so as to build a fast, efficient, and accurate dependency parsing model for professional domain texts.

## 3. Methods

### 3.1. Construction of Dependency Parsing Model with a Single Treebank

The dependency parsing model with a single treebank is the basis of the graph fusion model proposed in this paper. In this study, the maximum entropy model was used as both the dependency category and dependency probability estimation method. Based on the large-scale open news treebank and the professional element treebank, we established different dependency parsing models for the subsequent construction of graph fusion models. The tool used for constructing the news treebank and element treebank is the HIT-LTP [40] natural Language processing toolkit, which defines a total of 24 semantic dependency relations, and the subject, predicate, and object structures of sentences can be obtained by analyzing the dependency relations between words [41].

The construction of the dependency parsing model with a single treebank consists of two steps: (1) the construction of the input data instance and (2) the training of a maximum entropy model [42].

(1) Construction of input data instance

According to the definition of the maximum entropy model, the dependency syntactic treebank of the CoNLL scheme (Table 1) should be transformed into a binary group (<{features}, label>) composed of features and the output. For dependency syntax, “features” is the context information of a word, and “label” is the dependency tag.

Based on the dependency parsing theory of the maximum spanning tree, a word can have a dependency relationship with any word in a sentence theoretically. Therefore, *N*(*N* − 1) training examples will be generated for sentences with *N* words, which is too large for the model. In fact, for a long sentence, the possibility of dependency between words will decrease with the increased distance between words (the number of word intervals), especially when there is punctuation between words. According to the research [43], the number of dependencies with word distance within [−13, 15] accounts for 96.5% of the total number of dependencies. At this range of distance selection, a single word will produce 28 pairs of words at most. If there is a dependency between terms, then “label” is defined as the corresponding dependency category label as output; if there is no dependency category label, “null” will be used as the label.

In order to make full use of the annotation information provided in CoNLL, the vocabulary itself and the corresponding parts of speech should be involved in the construction of training cases. This paper uses “window” to effectively obtain the features between the dependent and the head. According to the sparseness of the word itself, we define windows of different sizes for the word itself and its part of speech. Considering the length of the Chinese phrase and the size of the training files, the size of the feature window of the word itself is set to 1, and the size of the part-of-speech window is set to 5.

Specifically, we utilize vocabulary, part of speech, and word distance to construct independent or combined features. For word pairs in positions *i* and *j* within a sentence (where ***i*** is the head and *j* is the dependent), *Word_i_* represents the word itself, corresponding to the ‘FORM’ column in Table 1. *Tag_i_* represents the word’s parts of speech, corresponding to the ‘UPOS’ column in Table 1. *Dis_i,j_* represents the distance between *Word_i_* and *Word_j_* (expressed in absolute value), and “+” represents the combination of features. The 15 features used in this paper are shown in Table 2.

In the feature value part of the feature table, the left side is the dominant feature, and the right side is the governed feature (except the distance feature). The features formed by them are expressed as “Dominator→Governed”. Taking the word “me” as an example from Table 1, its head is indexed as 2, indicating that the dominator word is “asked”, and the label corresponds to the value in the DEPREL column, which is “dobj.” The first feature pair is thus represented as (“asked”, “me”), the second feature pair is (“VERB”, “PRON”), and so on, with the 15th feature *Dis_i,j_* having a value of 1. Based on the dependency distance and feature table, several examples can be created for sentences with marked dependency syntactic relations.

(2) Training of maximum entropy model

Generalized Iterative Scaling (GIS) is an iterative optimization algorithm commonly employed for addressing maximum likelihood estimation problems, particularly in fields such as statistical modeling and natural language processing. GIS is renowned for its convergence properties and relative ease of implementation, which contribute to its outstanding performance and competitiveness in structured prediction tasks, such as dependency syntax analysis, especially in these domains. We use the training component provided by OpenNLP, and the GIS algorithm is used to train the maximum entropy model. GIS is an algorithm used in statistical modeling. It estimates model parameters by iteratively updating them based on observed and expected counts. It aims to maximize the likelihood of the data. The algorithm starts with initial parameter values and iteratively improves them. It computes expected counts in each iteration and adjusts the parameters to match the observed and expected counts. OpenNLP is a machine learning-based natural language processing toolkit under the Apache Foundation, which provides training components for maximum entropy and perceptrons, available at https://opennlp.apache.org/ accessed on 23 August 2023. The file structure of the maximum entropy model is shown in Table 3.

Firstly, according to the word pairs, their parts of speech and the distance between words, the features are established based on Table 2. Then, the dependency category and prior probability corresponding to each feature are obtained from the maximum entropy model. And the prior probabilities of each dependency category corresponding to all features are processed by parameter correction, logarithm, and normalization. Finally, the maximum probability value and the corresponding dependency category are selected as the link edge weight and dependency category label between the word pairs.

The algorithm for sentence parsing based on the single treebank dependency parsing model is shown in Algorithm 1.
**Algorithm 1:** Analysis method of single treebank dependency syntax parsing model.
Input: Sentence to be analyzed *Sentence*, Maximum entropy model *MaxEntModel*;
Output: Results of dependency parsing *TreeResult*;1Segmentation and part-of-speech tagging of *Sentence* and adding core nodes;2Constructing word pairs, CoWords = {(1,2), (1,3),…,(i,j),…,(n,n − 1)}(i≠j);3Initializing weight label and probability graph, Graph = {};4Foreach (i,j) in CoWords:{5    Initializing probability summation statistics, Weights = {0.0};6    Constructing the features of the vocabulary pair, Features(i,j) = {F_1_,…,F_15_};7    For each F_k_ in Features(i,j):{8        Get F_k_’s label from MaxEntModel *prior-labels*, probability *prior-weights*;9        Adding and update Weights, Weights[prior-labels] +=prior-weights;10        Removing the word pair from the word pair set, CoWords = CoWords-(i,j);}11        Parameter correction and logarithm, Weights = exp(Weights × Constant);12        Normalization processing, Weights = Normal(Weights);13        Taking out the maximum value and the corresponding category,         Graph_i,j_ = <Label, max(Weights)>;}14    Solving based on the maximum spanning tree of Graph;15Outputting the syntactic analysis result *TreeResult.*

### 3.2. Construction of Graph Fusion Model on Weight Optimization of Connected Edges

The public large-scale news treebank has a large number of natural language expression features, but it is common and not very professional; the sample size of the element treebank in the professional field is small, but it contains rich professional syntactic features. Therefore, this paper chooses a graph fusion model based on the weight optimization of the connection edges and makes full use of the rich corpus of the news treebank and the strong professionalism of the element treebank so as to establish a more accurate syntax tree. The basic idea of the graph fusion dependency parsing model is: according to the principle of the maximum spanning tree dependency parsing method [44,45] when constructing the weighted graph of the vocabulary pair, the graph fusion operation is introduced, that is, the one with higher dependency probability is selected from the parsing results of the two single treebank models. Dependencies are used as options for connecting edges in the graph to achieve model fusion optimization.

Firstly, two dependency parsing models based on a single treebank are constructed by using the news treebank and the element treebank, respectively, with the method in Section 3.1. Secondly, according to the sentence and its segmentation results, the features of the word pairs are established. And the news model and the transcript model are mapped, respectively, to statistically solve the set of labels and probability pairs of output categories. Let the news corpus set be ***S**_g_***, the professional corpus set be *S_b_*, and the label and probability pair are expressed as <*L_x,i_*, *W_x,i_*>(*x*∈{*g, b*}, *i*∈[1,n]). The set is shown in Formula (1), and the maximum probability obtained from the set is shown in Formula (2), where *W_x,max_*(*x*∈{*g, b*}) represents the maximum value and *S_x,values_*(*x*∈{*g, b*}) represents the value of the key–value pair.
(1)$$S_{g}=\left\{\langle L_{g,l},W_{g,l}\rangle ,\cdots ,\langle L_{g,n},W_{g,n}\rangle \right\}\;S_{b}=\left\{\langle L_{b,l},W_{b,l}\rangle ,\cdots ,\langle L_{b,n},W_{b,n}\rangle \right\}$$
(2)$$W_{g,max}=argmax(S_{g,values})\;W_{b,max}=argmax(S_{b,values})$$

Then, the graph fusion operation is performed; that is, the larger of them is selected as the category and possibility result of the connected edge, and the edge *W(i,j)* is assigned a value, as shown in Formula (3).
(3)$$W\left(i,j\right)=\left\{\begin{array}{c} \langle L_{g,max},W_{g,max}\rangle ,if\;W_{g,max}>W_{b,max}\\ \langle L_{b,max},W_{b,max}\rangle ,if\;W_{g,max}\leq W_{b,max} \end{array}\right.$$

Finally, the dependency syntax tree results are obtained from the fused graph based on the maximum spanning tree algorithm. The flow of the graph fusion model to realize dependency parsing is shown in Figure 2.

The analysis process of the graph fusion model proposed in this paper is described in Algorithm 2.
**Algorithm 2:** Steps of graph fusion dependency syntax parsing model.
Input: sentence to be analyzed *Sentence,* Transcript model *CaseRecordModel*, News model *NewsModel*
Output: Dependency parsing results *TreeResult*1Segmentation and part-of-speech tagging of *Sentence* and adding core nodes;2Constructing word pairs, CoWords = {(1,2), (1,3), …(i,j)…, (n,n − 1)}(i≠j);3Initializing weight label and probability graph, Graph = {};4Foreach (i,j) in CoWords: {5    Initializing probability summation statistics, WeightsCR = {0.0}, WeightsN = {0.0};6    Construct the features of the vocabulary pair, Features(i,j) = {F_1_,…,F_15_};7    For each F_k_ in Features(i,j):{8        Get prior_labels_CR and prior_weights_CR according to CaseRecordModel;9        Get prior_labels_N and prior_weights_N according to NewsModel;10        Update WeightsCR: WeightsCR[prior_labels_CR] += prior_weights_CR;11        Update WeightsN: WeightsN[prior_labels_N] += dprior_weights_N;12        Remove the vocabulary pair from the vocabulary pair set: CoWords = CoWords-(i,j);13        Parameter correction and logarithmization of probability,        WeightsCR = exp(WeightsCR × Constant), WeightsN = exp(WeightsN × Constant);14        Normalization: WeightsCR = Normal(WeightsCR), WeightsN = Normal(WeightsN);}15    Get the maximum weight and dependency label    <L_CRmax,W_CRmax>, <L_Nmax,W_Nmax>;16    If W_Crmax ≥ W_Nmax:17        Graphi,j ≤ L_CRmax,W_CRmax>;18    Else:19        Graphi,j ≤ L_Nmax,W_Nmax>;}20Solving Maximum Spanning Tree Based on Graph;21Output the dependency parsing result *TreeResult.*

### 3.3. Generation of DAT-MT High-Performance Syntax Tree

The graph fusion dependency syntactic analysis model proposed in this paper increases the amount of model calculation steps, and the performance decreases. As a result, the graph fusion model cannot meet the high-performance computing requirements of real-time processing and quality inspection on the professional corpus. Therefore, it is necessary to rely on graph fusion to improve the parsing speed of the syntactic model. In this regard, this paper will improve performance from two aspects: establishing efficient indexing of data and designing parallel computing methods. Considering the large difference in scale between the news model and the element model [46,47,48,49], the DAT-MT model is proposed. As a data structure of the model feature index, DAT can make full use of computing resources and reduce the waiting time consumption of synchronization suspension during parallel computing [50]. At the same time, DAT uses double arrays to effectively compress the storage space, achieving the effect of high query efficiency and low storage occupation, and with DAT, time complexity does not change with the data volume. In terms of multi-threading, estimating the weights of each connection edge in the graph fusion model is the most time-consuming step. Additionally, the calculation of dependency probability and dependency category between each word pair is independent. Therefore, we design a parallel process of connecting edge weight calculation to achieve efficient model calculation.

The DAT-MT high-performance syntax tree generation method uses the DAT to organize the model data and uses multi-threads as the model analysis calculation method to realize the high-performance generation of a syntax tree. The principle of acceleration is shown in Figure 3.

(1) The use of DAT

DAT-MT acceleration method realizes the efficient indexing and mapping of model features to prior parameters with high efficiency, equal time complexity, and low storage consumption by utilizing DAT. As shown in Figure 3 ①, firstly, the prior parameters are stored in the two-dimensional array according to the model, where the subscript of the array indicates the position of the prior parameters in the two-dimensional array. Secondly, the key–value pair < Feature, Order > is formed by the feature and the array subscript of the corresponding prior parameter so as to construct the parameter mapping from the feature to the prior parameter, and the key–value pair is inserted into DAT with the feature character as the index value to complete the establishment of the index structure. Finally, the news model and the element model are indexed and mapped by this method and loaded into the memory for the query.

DAT uses two arrays, base and check, to complete the expression of deterministic finite automata, where the elements in the base array are used to express the nodes in the Trie tree, that is, a state. The check array represents the previous state of a certain state. In the initial state, both base and check are 0. When constructing, if the corresponding state is a complete word, the base value is set to a negative number. If a complete word is formed in a certain state, but the word is not the prefix of other words, its base value can take the negative number of its state position.

If the input character is defined as c and the state is changed from *S_m_* to *S_n_*, the change in the two arrays can be expressed as Formula (4).
(4)$$base\left(S_{m}\right)+c=base\left(S_{n}\right)\;check\left(S_{n}\right)=S_{m}$$

Because the DAT model has high query efficiency and time complexity only related to character length, it can reduce the time cost of thread synchronization. For the query of a certain feature, it is only necessary to search the feature to be queried in DAT, take out the subscript Order of the parameter accordingly, and then obtain the prior parameter from the corresponding position in the two-dimensional array where the parameter is stored with the time complexity of O(1). The query time of the DAT data structure is only related to the characteristic string length, which is O(n) (n is the string length). Therefore, for the same feature, although the sizes of the news model and transcript model are quite different, the query time under the model file organization method established in this study is O(1) + O(n), which can achieve a high degree of parallelism.

(2) The use of MT

The DAT-MT acceleration method uses the Fork–Join parallel computing model to decompose and parallelize the syntax tree generation steps based on establishing the DAT index for model files. The Fork–Join model is a classic shared-memory multi-threading parallel model [51]. In this model, all programs start with a single main thread, which creates multiple threads to enter parallel regions for concurrent computation. When the tasks in the parallel region are completed, the child threads synchronize, allowing the program to continue under the control of the main thread until the entire computational task is finished. The basic idea is that for the edge weight estimation step, which takes the most time to calculate and the calculation of each edge is independent, the process is put into the parallel interval. As shown in Figure 3 ②, it is divided into three stages.

In the first stage, same as the RB-ST (red-black tree index and single-thread calculation), the DAT-MT method starts with a main thread, constructs the words contained in the sentence into word pairs, and initializes the matrix graph accordingly to store the weighted graph. DAT-MT puts the weighted graph to be constructed into the shared memory area for each thread to assign values [52].

The second stage is the Fork stage. The DAT-MT method takes the decomposed vocabulary pairs as independent computing units, enters the parallel interval, and solves the dependency relationship and the existence probability of the dependency relationship between the vocabulary pairs in parallel. Specifically, after completing the word segmentation processing and the construction of the vocabulary pairs, firstly, according to the independence of each weight calculation, it decomposes a plurality of threads and realizes the independent construction of features in each thread. Then, aiming at the characteristic that the feature retrieval time consumption is only related to the character length of the news model and the professional field element model with DAT, the parameter analysis is carried out in parallel. Finally, the connection edge weight optimization operation based on the graph fusion method is carried out to complete the weight estimation and dependency category selection, and the matrix graph is assigned values according to the parameter estimation results until all the word pairs are calculated in the parallel interval.

In the third stage, Join, we solve the maximum spanning tree of the assigned graph matrix and output the dependency parsing result of the sentence; after all the word pairs are solved in parallel, it returns to the main thread.

## 4. Results

### 4.1. Introduction of Experiment

In order to verify the effect of improving the accuracy and speed of our model, this experiment selects the professional field of telecommunication fraud record analysis. The news data were sourced from publicly available People’s Daily corpora, which can be openly accessed at https://opendata.pku.edu.cn/dataverse/icl accessed on 23 August 2023. Following the methodology outlined in Section 3.1, we established a dependency syntactic analysis model for the news corpus, referred to as the “news model”. The telecommunication fraud transcript data was supplied by our project’s collaborative partner. It documents the dialogues between law enforcement officers and the victims, encompassing a comprehensive range of case-specific information, including details pertaining to the victims, the sequence of events in the fraudulent activities, and information regarding the suspects, as illustrated in Figure 4.

For the telecommunication fraud transcript data, we annotated dependency syntactic relationships for a total of 413 sentences. We divided this dataset into training and test sets following a 3:1 ratio. We use the training set of the telecommunication fraud element tree as the corpus to construct a domain-specific element model for the field of public security, specifically the transcript model. Then, we use the news model and the transcript model as the basic model to build a graph fusion dependency parsing model for the text of telecommunications fraud transcripts. The experiment takes the test set of the telecommunication fraud element tree library as the object of syntax analysis and compares the syntactic analysis results obtained by the graph fusion model proposed in this paper with the news model, transcript model, and open-source Chinese syntactic analysis tools.

### 4.2. Experimental Results

#### 4.2.1. Dependency Syntax Tree Extraction

The evaluation of the dependency parsing model can be divided into two aspects: the accuracy of the selection of dependent word pairs and the accuracy of the judgment of dependency categories. The Unlabeled Attachment Score (UAS) is used to evaluate the accuracy of the selection of dependent word pairs. It is defined as the percentage of all words for which the correct dominant word is found among all words containing the root node (predicate verb). Labeled Attachment Score (LAS) is used to evaluate the dependency category judgment accuracy rate, which is defined as the percentage of all words that contain the root node (predicate verb) for which the correct dominant word is found and the class of dependencies between its dominant and dominated is predicted correctly. According to the above definition, let the total number of words be sum, the number of correct dominant words is ${num}_{UnLabel}$, and the number of words whose correct dominant words and dependent categories are both correct is ${num}_{Label}$; then, the calculation method of UAS and LAS is shown in Formula (5).
(5)$$UAS=\frac{{num}_{UnLabel}}{sum}*100\%\;LAS=\frac{{num}_{Label}}{sum}*100\%$$

In addition, this paper uses syntactic analysis speed (Speed) to evaluate the syntactic analysis speed performance of the model, which is defined as the number of sentences parsed by the model per second (including the model loading time).

We use UAS, LAS, and Speed to evaluate the accuracy and parsing efficiency of the dependency parsing model and compare it with the dependency parsing model based on deep learning. Among them, DDParser is a dependency parsing tool based on deep learning platform PaddlePaddle and large-scale labeled data training [53], available at https://github.com/baidu/DDParser accessed on 23 August 2023; LTP4 is a deep learning model based on PyTorch [54], which provides Chinese natural language processing tools including syntax analysis, and is available at https://github.com/HIT-SCIR/ltp accessed on 23 August 2023. The obtained results are shown in Table 4.

In terms of the accuracy of the dependency parsing model, by comparing the UAS and LAS of the graph fusion model proposed in this paper with the news model and transcript model, we can see that the UAS and LAS of the graph fusion model were greatly improved by 13.34% and 14.82%, respectively compared with the transcript model, and by 3.14% and 3.40%, respectively compared with the news model. The accuracy of the news model is acceptable, while the accuracy of the transcription model is relatively low. This is because the transcription model is trained using a smaller-scale constituent sentence treebank, which results in an insufficient number of input instances and an incomplete representation of syntactic relationships. Consequently, it is unable to learn comprehensive dependency relationships and dependency categories, highlighting the inability of small-scale data to independently construct dependency parsing models and demonstrating the necessity of training on large-scale treebanks. When compared to DDParser and LTP4, it can be observed that the graph fusion model shows some improvement in UAS, with increases of 13.14% and 9.17%, respectively. Furthermore, there is a significant improvement in LAS, with increases of 13.54% and 20.11%, respectively. DDParser and LTP4 are deep learning models trained on large-scale corpora, which achieve high accuracy in predicting dependency relationships. However, these models lack the knowledge of specialized domains, and when confronted with colloquial and specialized syntax found in telecommunication fraud transcriptions, they are unable to effectively identify the dependency categories, resulting in lower LAS. The graph fusion model can learn both the general patterns from large-scale corpora and the specific grammar characteristics of transcription data, leading to improved reliability in dependency parsing trees. 

However, for the dependency parsing speed, the analysis speed of the transcript model using only the element sentence tree database is about 330 sentences per second due to the small scale of the model, the analysis speed of the news model is 56.4 sentences, and the syntax analysis speed of the graph fusion model is only 51.5 sentences per second, while the syntactic analysis speed of DDParser and LTP4 based on deep learning is slower, respectively 29.0 sentences per second and 12.1 sentences per second, which reflects the necessity of designing efficient parallel algorithms.

#### 4.2.2. Performance Optimization of the Model

Firstly, the advantages of double-array Trie in model data organization, from feature to parameter mapping efficiency, storage space occupancy, and query time stability are verified. In this paper, we compare the performance of DAT with the B+ tree, red-black (RB) tree, and Trie tree under different model feature string data volumes with query time (per million times/ms) and index structure memory consumption (MB) as evaluation indexes.

Features of 100K, 200K, 500K, 1M, and 3M were successively selected from the news model as the searched set, and 1K features were randomly selected as the search object for the experiment. In this study, experimental results were obtained under the computing environment of CPU: 1.8 ghz; memory: 8 G; and operating system: Win10-64-bit, as shown in Figure 5 and Figure 6, respectively.

Figure 5 shows that the query time of the Trie tree and its DAT variant is less than that of the other two search tree structures under different feature numbers, and the query efficiency ratio is about 2.4 times higher than that of the red-black tree when the feature number is 3M. In addition, with the increase in data volume, the query efficiency of the two remains stable, and the difference in query time per million is less than 30 ms for different feature numbers, which verifies that this type of data structure has the advantages of efficient and time-consuming mapping from feature strings to prior parameters under different scale dependency parsing model. Figure 6 shows that according to the memory footprint of Trie and its DAT variant, it can be found that the memory footprint of the Trie tree increases rapidly as the number of features increases. When the number of features is one million, the memory footprint will reach the GB level, which will have a negative impact on the system’s performance. However, the memory footprint of DAT is relatively low. With the same number of features, the memory footprint of DAT is only about a quarter of that of naive Trie, which verifies the good space compression performance of DAT. To sum up, the experiment proves that DAT has the advantages of high indexing efficiency of Chinese character strings, low occupancy rate of system resources such as memory, and the advantage that the query time complexity is only related to the length of character strings, but not related to the size of data, which can realize fast mapping of features to parameters and meet the parallel parsing requirements of graph fusion dependency parsing model.

Then, the validity of the DAT-MT acceleration method in dependency syntax tree generation is verified. In this study, the number of sentences parsed by the model per second is taken as the evaluation index, and the simulation experiment of syntactic analysis of sentences with different lengths (including 10, 20, and 30 words) by the graph fusion model is carried out. DAT-MT method is compared with red-black tree serial (RB-ST), DAT serial (DAT-ST), and red-black tree multi-threading (RB-MT), respectively, in the experiment. The results obtained are shown in Figure 7 (CPU: 4 × 1.8 ghz; memory: 8 G; operating system: Win10-64-bit).

According to Figure 7, using DAT as the model index has a small improvement in parsing efficiency compared with using RB as the index, which further verifies that DAT as the model feature-to-parameter mapping method has the advantages of high query efficiency and low synchronization cost. Compared with the single-thread or multi-thread calculation methods (RB-ST and RB-MT; DAT-ST and DAT-MT) under the same index method, it can be concluded that the multi-thread calculation method is several times faster due to the parallelization of model parsing. In general, when sentences contain 10, 20, and 30 words, the DAT-MT method achieves 3.77, 3.76, and 3.73 times of acceleration compared with RB-ST method, respectively, which demonstrates the effectiveness of the DAT-MT method.

Considering the phenomenon that the acceleration ratio of DAT-MT to RB-ST decreases with the increase in the number of words contained in the sentence, combined with the thread parallel strategy, it can be analyzed that the synchronization cost of long sentences increases because the graph needs to be assigned after the calculation of a single connected edge, but in general, the acceleration ratio is stable, which verifies the rationality of dividing parallel and serial intervals in this paper. In summary, the DAT-MT method proposed in this paper can effectively improve the parsing efficiency of the graph fusion model, thereby meeting the high-performance computing requirements of dependency parsing. Considering the phenomenon that the acceleration ratio of DAT-MT to RB-ST decreases with the increase in the number of words contained in the sentence, combined with the thread parallel strategy, it can be analyzed that the synchronization cost of long sentences increases because the graph needs to be assigned after the calculation of a single connected edge, but in general, the acceleration ratio is stable, which verifies the rationality of dividing parallel and serial intervals in this paper. In summary, the DAT-MT method proposed in this paper can effectively improve the parsing efficiency of the graph fusion model, thereby meeting the high-performance computing requirements of dependency parsing. The DAT-MT, on the one hand, combined with the characteristics of high retrieval efficiency and stable query time of DAT, realizes the efficient operation and synchronous simplification of the parsing of treebank dependency parsing models of different scales. On the other hand, it makes full use of multi-core computing so that the resource implements the parallel processing of syntax tree generation.

## 5. Conclusions and Future Work

Aiming at the specialization of language expression and the small sample size of the corpus in specific fields, this paper proposes the DAT-MT-accelerated graph fusion dependency parsing model. The graph fusion method based on connection edge weight optimization integrates the specialized syntactic features contained in a small amount of professional element treebank into the news model constructed from a public large-scale annotated news text corpus. The experiments show that the performance of the graph fusion model in the element sentence tree database test set is significantly better than that of the single tree database model and the deep learning-based dependency parsing model; at the same time, in response to the analysis and processing requirements of massive text data, the DAT-MT acceleration method is proposed. The DAT-MT acceleration method proposed in this paper can achieve several times the acceleration ratio compared with the traditional red-black tree index and single-threaded computing method.

Indeed, there is still room for improvement in the current study. In the feature selection of the graph fusion dependency parsing model, the effect of the model can be further improved by changing the window size of lexical feature selection or adding new combined features to build more effective features. In the estimation of edge weights, multiple types of machine learning models are utilized and fused via data normalization and other processing methods, thereby further improving the accuracy of the fusion model.

In addition, for the downstream tasks of text information processing, the DAT-MT accelerated graph fusion dependency parsing model can provide a reliable dependency syntax tree for the extraction of professional domain elements based on dependency syntax on one hand. On the other hand, it also provides high-performance processing support for text data for the construction of knowledge graphs in professional fields.

## Figures and Tables

**Figure 1 entropy-25-01444-f001:**
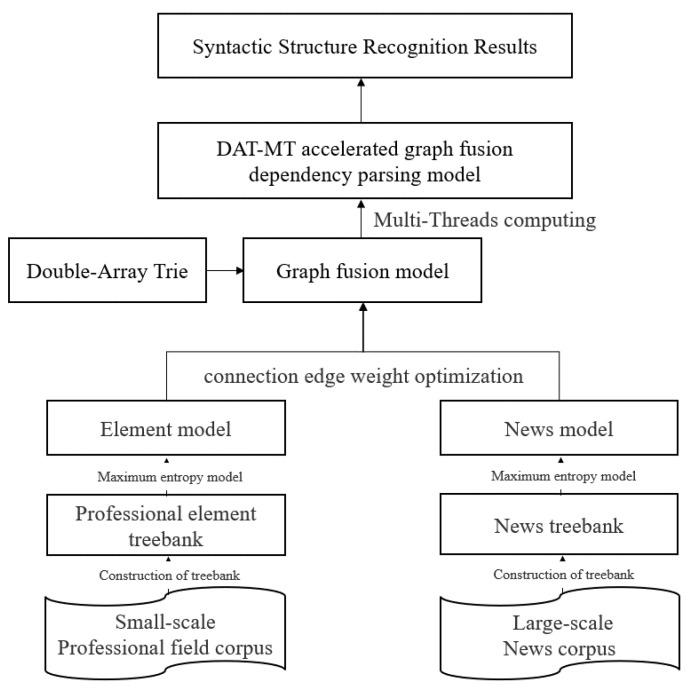
The flowchart of the DAT-MT accelerated graph fusion dependency parsing model.

**Figure 2 entropy-25-01444-f002:**
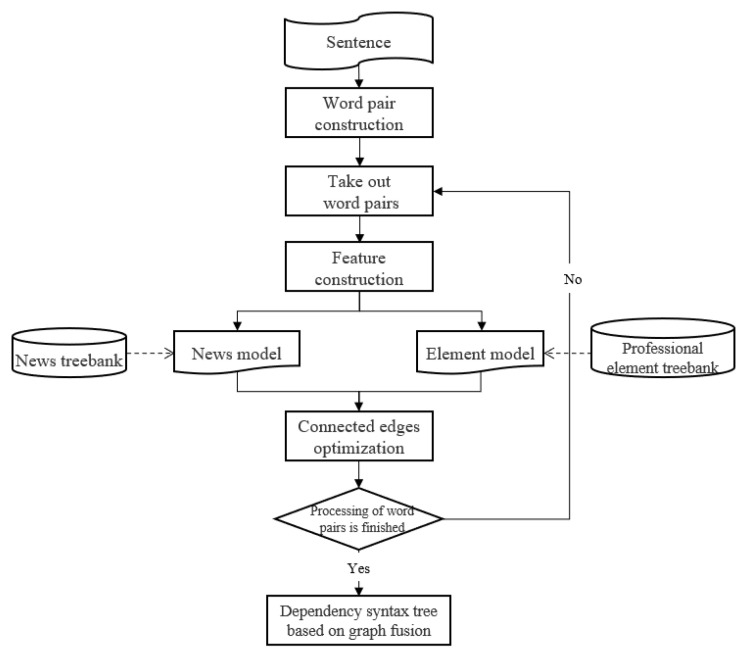
The flow chart of the fusion dependency parsing model.

**Figure 3 entropy-25-01444-f003:**
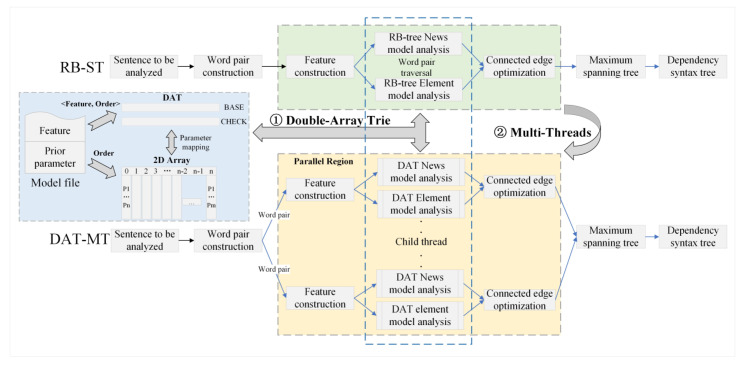
DAT-MT acceleration schematic.

**Figure 4 entropy-25-01444-f004:**
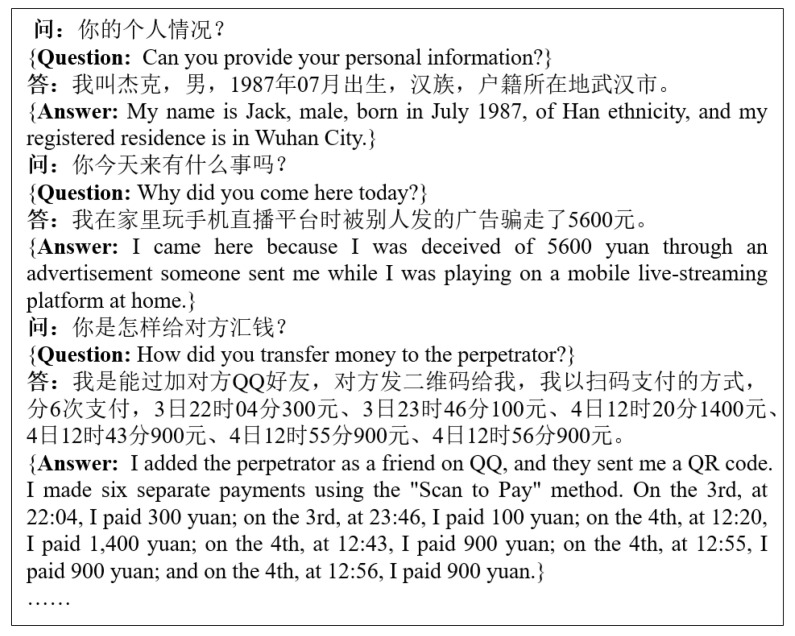
Example of a telecommunications fraud transcript file.

**Figure 5 entropy-25-01444-f005:**
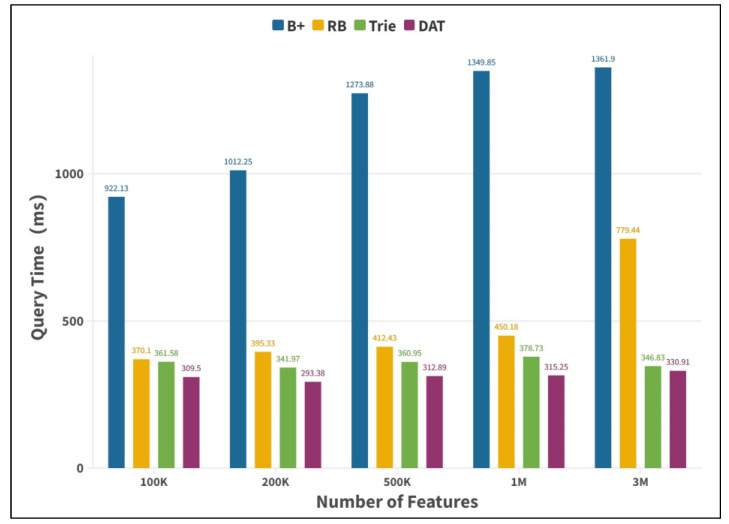
Tree query time consumption comparison histogram.

**Figure 6 entropy-25-01444-f006:**
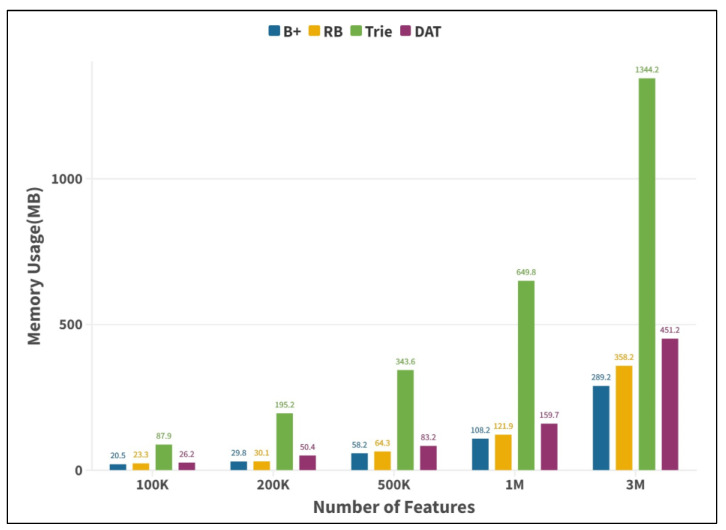
Tree memory consumption comparison histogram.

**Figure 7 entropy-25-01444-f007:**
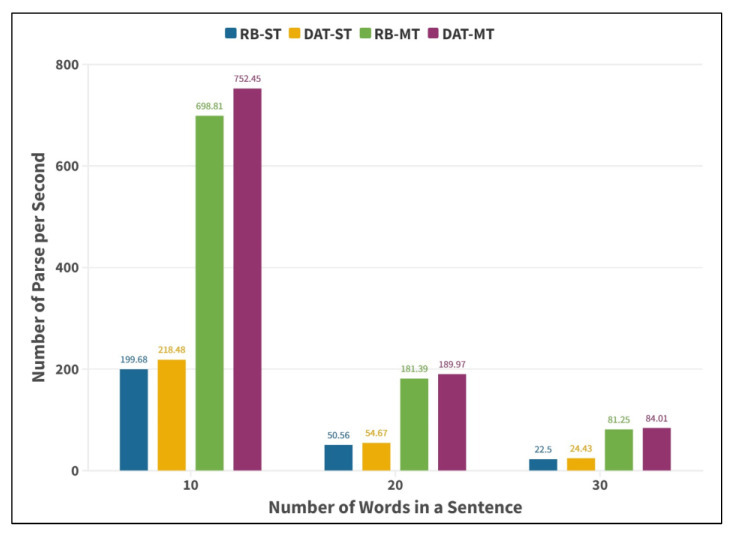
Comparison of dependency syntax parsing efficiency.

**Table 1 entropy-25-01444-t001:** The format of CoNLL.

ID	FORM	UPOS	XPOS	FEATS	HEAD	DEPREL
1	He	PRON	PRP	-	2	nsubj
2	asked	VERB	VBD	-	0	ROOT
3	me	PRON	PRP	-	2	dobj
4	to	PART	TO	-	5	aux
5	pay	VERB	VB	-	2	xcomp
6	1500	NUM	CD	-	7	nummod
7	yuan	NOUN	NN	-	5	dobj
8	to	PART	TO	-	9	aux
9	start	VERB	VB	-	5	advcl
10	the	DET	DT	-	13	det
11	automatic	ADJ	JJ	-	13	amod
12	delivery	NOUN	NN	-	13	compound
13	function	NOUN	NN	-	9	dobj
14	.	PUNCT	.	-	2	punct

**Table 2 entropy-25-01444-t002:** Dependency context feature table.

ID	Feature Value (Dominator on the Left, Governed on the Right)	
	Dominator	Governed
1	*Word_i_*	*Word_j_*
2	*Tag_i_*	*Tag_j_*
3	*Tag_i_* _−1_	*Tag_j_* _−1_
4	*Tag_i_* _+1_	*Tag_j_* _+1_
5	*Tag_i_* _+2_	*Tag_j_* _+2_
6	*Tag_i_* _−2_	*Tag_j_* _−2_
7	*Tag_i_* + *Tag_j_* + *Tag_i_*_+1_	*Tag_i_* + *Tag_j_* + *Tag_j_*_+1_
8	*Tag_i_* + *Tag_j_* + *Tag_i_*_−1_	*Tag_i_* + *Tag_j_* + *Tag_j_*_−1_
9	*Tag_i_*_+1_ + *Tag_i_*_+2_	*Tag_j_*_+1_ + *Tag_j_*_+2_
10	*Tag_i_*_−1_ + *Tag_i_*_−2_	*Tag_j_*_−1_ + *Tag_j_*_−2_
11	*Word_i_* + *Tag_i_*	*Word_j_* + *Tag_j_*
12	*Word_i_* + *Word_j_*	*Tag_i_* + *Tag_j_*
13	*Word_i_* + *Tag_i_* + *Dis_i,j_*	*Word_j_* + *Tag_j_* + *Dis_i,j_*
14	*Word_i_* + *Word_j_* + *Dis_i,j_*	*Tag_i_* + *Tag_j_* + *Dis_i,j_*
15	*Dis_i,j_*	

**Table 3 entropy-25-01444-t003:** The file structure of OpenNLP maximum entropy GIS model.

Storage Content	Model Data Instance	Number of Rows
Model Category Description	GIS	1
Correction Constant	1	1
Calibration Parameter	0.0	1
Number of Dependent Categories	16	1
Dependent Category Label	SBV	number of dependent categories
Number of Patterns	12,969	1
Pattern Details	2 0 1 2 3 4 5 6 7 8 9	same as number of patterns
Number of Features	3,187,863	1
Feature Details	v→n	same as number of features
Prior Parameter	0.18020979088959754	number of features × corresponding number of patterns

**Table 4 entropy-25-01444-t004:** Test result table of dependency parsing model.

Indicator	News Model	Transcript Model	DDParser	LTP 4	Graph Fusion Model
UAS (%)	76.55	66.35	66.55	70.52	79.69
LAS (%)	73.06	61.64	62.92	56.35	76.46
Speed (number of sentences/second)	56.4	330.1	29.07	12.1	51.5

## Data Availability

The data that support the findings of this study are available from the first author, Li, R., upon reasonable request.

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
