# Peer review of "DAT-MT Accelerated Graph Fusion Dependency Parsing Model for Small Samples in Professional Fields"

_entropy, 2023, doi:10.3390/e25101444_

Round 1
Reviewer 1 Report
The paper proposes a Double-Array Trie and Multi-threading (DAT-MT) accelerated graph fusion dependency parsing model for understanding the domain migration of syntactic tree libraries and improving the speed of text parsing. It combines the syntactic and other general features to improve the syntactic relation extraction. I found the work to be interesting and I didn't have many questions.
The authors have presented an interesting work. The manuscript is clear and well presented. The authors have conducted acceptable experiments and comparisons.
Topic is relevant in the field. It is an interesting work.
Double-Array Trie and Multi-threading (DAT-MT) accelerated graph fusion dependency parsing model seems to be promising and new when compared to the existing works. The authors can present this more clearly, if required.
References look appropriate.
Minor comments:
1. The clarity of figures is poor. The authors should present high quality figures.
Reviewer 2 Report
The paper introduces an innovative method for dependency parsing tailored to specific domains, demonstrating significant advancements in both accuracy and processing speed. The authors achieve accuracy enhancements by integrating generic and domain-specific dependency parsing models, while processing speed gains are realized through the implementation of a multi-threaded Double-Array Trie structure. The empirical results clearly indicate the superiority of this approach compared to existing models.
Nevertheless, the manuscript could benefit from further elucidation in the following areas:
Feature Encoding (Table 2): The authors should provide a more detailed account of how each word and tag is encoded. Specifically, it would be valuable to clarify whether the authors employ a Bag-of-Words (BOW) implementation or employ a word embedding approach. A comprehensive explanation of this preprocessing step would greatly enhance the paper's clarity and assist readers in better understanding the methodology.
Choice of Generalized Iterative Scaling (GIS): The authors have chosen Generalized Iterative Scaling (GIS) as their training algorithm, but the rationale behind this selection remains unclear. It would be advantageous to provide a justification for why GIS was chosen over other machine learning algorithms, and whether any comparative experiments or empirical evidence support this choice.
Test Dataset Origin: The paper references a test dataset for model validation, but it is ambiguous whether this dataset is derived solely from the telecommunication fraud dataset, or if it is a composite dataset comprising both the telecommunication fraud and news datasets. Clarifying the exact origin of the test dataset will enhance the reproducibility and transparency of the research.
Incorporating these clarifications and explanations will further enrich the quality and comprehensibility of the paper, ensuring that readers have a more comprehensive understanding of the methodology and its contributions.
Reviewer 3 Report
p.3 DAT-MT is the proposed method, so a phrase 'a DAT-MT' is meaningless. Unless there is an existing method that has been modified to create this one, in which case it needs to be described in detail and its differences from the proposed one should be provided in method's description.
Tables 3&4 They are hard to read because text in them is not left-aligned.
Algorithm 2: Same comment. It is impossible to read.
p.9 The DAT model is nothing new, it needs to be described with appropriate references:
Aoe, J.I., Morimoto, K. and Sato, T., 1992. An efficient implementation of trie structures. Software: Practice and Experience, 22(9), pp.695-721.
Then, the authors need to explain how does their modification improve the model and to justify this modification. Does it improve scores only, runtime, or both? If so, to what extent?
p.10 Again, DAT appears here without a reference, and it is not an original contribution of this paper.
p.10 "DAT model has high query efficiency and time complexity only related " - compared to what? Why? Where is a justification, experimental or referential, of that claim?
p.10 "Fork-Join parallel computing model " - again, no citation and no description.
p.11 "To sum up, the DAT-MT, on the one hand, combined with the characteristics of high retrieval efficiency and stable query time of DAT, realizes the efficient operation and synchronous simplification of the parsing of treebank dependency parsing models of different scales. " - there is no proof to that claim. If a new algorithm is suggested, then complexity analysis should be given, together with complexities of competing methods that are not mentioned here.
p.12 No citations for other parsers are provided.
p.12 No data stats or description is provided. On what data Table 4 results hold???
I could not read the paper further because it is impossible to understand what is tested on what here and how to interpret the results.
English is not a big problem here, but the presentation is. Tables, algorithms, and formulae are very hard to read due to wrong alignment and lack of top and bottom space around the equations. The authors make claims that are not supported by experimental evaluation OR theoretical analysis.
Round 2
Reviewer 3 Report
All of my comments were adequately addressed. The manuscript was improved sufficiently.